# Effects of larval rearing substrates on some life-table parameters of *Lutzomyia longipalpis* sand flies

Kelsilandia Aguiar Martins[1¤]*, Maria Helena de Athayde Meirelles[1,2], Tiago Feitosa Mota[1], Ibrahim Abbasi[3], Artur Trancoso Lopo de Queiroz[1], Claudia Ida Brodskyn[1,2], Patrícia Sampaio Tavares Veras[1], Deborah Bittencourt Mothé Fraga[1,2‡], Alon Warburg[3‡]

**1** Instituto Gonçalo Moniz-Fundação Oswaldo Cruz, Salvador, Brazil, **2** Escola de Medicina Veterinária e Zootecnia-Universidade Federal da Bahia, Salvador, Brazil, **3** Kuvin Center for the Study of Infectious & Tropical Diseases, Department of Microbiology and Molecular Genetics, Institute of Medical Research, Israel-Canada, Faculty of Medicine, The Hebrew University of Jerusalem, Jerusalem, Israel

¤ Current address: The Royal Veterinary College, University of London, Hawkshead Lane, London, United Kingdom
‡ These authors contributed equally as senior authors.
* kaguiarmartins@rvc.ac.uk

**Data Availability Statement:** All relevant data are within the manuscript and its Supporting Information files.

## Abstract

Sand flies are the insects responsible for transmitting *Leishmania* parasites, the causative agents of leishmaniasis in humans. However, the effects of sand fly breeding sites on their biology and ecology remain poorly understood. Herein, we studied how larval nutrition associated with putative breeding sites of the sand fly *Lutzomyia longipalpis* affects their oviposition, development, microbiome, and susceptibility to *Leishmania* by rearing *L. longipalpis* on substrates collected from an endemic area for leishmaniasis in Brazil. The results showed that female *L. longipalpis* select the oviposition site based on its potential to promote larval maturation and while composting cashew leaf litter hindered the development, larvae reared on chicken feces developed rapidly. Typical gut microbial profiles were found in larvae reared upon cashew leaf litter. Adult females from larvae reared on substrate collected in chicken coops were infected with *Leishmania infantum*, indicating that they were highly susceptible to the parasite. In conclusion, the larval breeding sites can exert an important role in the epidemiology of leishmaniasis.

## Author summary

Sand flies are the insect vectors involved in the transmission of many pathogens, however, the transmission of parasites to humans leading to visceral leishmaniasis is currently the most critical threat caused by this insect. Despite the importance of the vector, many aspects of the biology of sand flies are poorly understood, especially their breeding sites. This study was designed to evaluate the oviposition, life span, microbiome, and parasite infections in the main species of sand fly responsible for visceral leishmaniasis in America. Insects were reared on substrates collected from different putative habitats of sand flies in

**Funding:** This study was supported by Conselho Nacional de Desenvolvimento Científico e Tecnológico-CNPq https://www.gov.br/cnpq/pt-br, Grant Number: PVE 401213/2014-5 and 305235 / 2019-2) received by PSTV and Grant Number: 304876/2019-4 by CIB. It was also supported by Fundação de Amparo à Pesquisa do Estado da Bahia- FAPESB http://www.fapesb.ba.gov.br/, Grant Number: FAPESB 04/2013, 12/2014 and 04/ 2015 received by CIB. AW received funding from The Israel Science Foundation-https://www.isf.org. il/#/- (Grant Number: 997/19). The funders had no role in study design, data collection and analysis, decision to publish, or preparation of the manuscript.

**Competing interests:** The authors have declared that no competing interests exist.

an endemic area for the disease in Brazil. The results showed that female vectors selected an oviposition site depending on the potential offered to their offspring. Furthermore, the development of immature stages varied according to the type of substrate evaluated, with cashew leaves litter delaying larval development, while chicken shelter promoted larval development. The challenge of females emerging from chicken shelter substrate with the parasite indicates that insects reared in such an environment could successfully sustain the infection. These results suggest that the type of breeding site can affect insect biology as well as the epidemiology of the disease.

## Introduction

Phlebotomine sand flies (Diptera: Psychodidae: Phlebotominae) are hematophagous insects of great medical and veterinary importance due to their ability to transmit bacterial, viral, and protozoan pathogens [1]. Particular attention is accorded to the transmission of protozoans of the genus *Leishmania* (Kinetoplastida: Trypanosomatidae) since several species of sand flies are involved in the transmission of these parasites to humans [2]. Zoonotic transmission of *Leishmania* is facilitated by female sand flies who feed on infected hosts. Infections with different *Leishmania* species manifest locally at the bite site causing cutaneous leishmaniasis, destroying mucous membranes during mucocutaneous leishmaniasis, or metastasize to internal organs, such as the spleen, liver, and bone marrow, causing life-threatening visceral leishmaniasis (VL) [3]. Every year 1–2 million people are diagnosed with leishmaniasis, ranking the disease as one of the most important neglected tropical diseases [4]. American VL is caused by infection with *Leishmania infantum*, in which domestic dogs serve as the reservoir hosts [5]. In South America, the disease is endemic to twelve countries, with 96% of the cases diagnosed in Brazil [6].

*Lutzomyia longipalpis* is the most important vector of VL in the Americas [7,8]. It is a species complex that originates in forest habitats where sand flies used to feed exclusively on wild animals and plants [9]. However, with the encroachment of human habitation on sylvatic habitats, *L. longipalpis* proved exceptionally adaptable to peri-domestic rural habitats, shifting its blood-feeding preferences to domestic animals and humans [10]. Currently, VL in Brazil is becoming increasingly urbanized with transmission occurring in major cities [11,12]. Since *L. longipalpis* is by far the most critical vector of VL, curtailing its population is a common approach to reduce the morbidity caused by VL, typically by residual insecticide spraying with pyrethroids targeting adult sand flies resting on walls and fences [13,14].

Unlike mosquitoes, the immature stages of sand flies develop in terrestrial habitats with high humidity that are rich in decaying organic matter or humus. Once the eggs hatch, larvae undergo four instars and pupate, with the entire lifecycle, including adults, lasting approximately 1–2 months [15]. Sand fly breeding sites could be an alternative target for control since the immature stages are slow and comprise the longest phase of the life cycle. However, breeding sites are hard to identify as they can occupy a wide range of environments [16,17]. Therefore, identifying the breeding sites for immature stages remains one of the most difficult challenges in the research of these insects.

Previous lab and fieldwork have identified environmental parameters related to the development of these insects, typically, the immature stages require high humidity, organic matter, and darkness [18,19]. More recently, the microbiota has been explored as another important environmental factor affecting sand fly biology. The presence of certain microorganisms can attract gravid females to appropriate breeding sites [20]. Moreover, the microbiome of larvae

is essential for proper development and can influence adults' susceptibility to *Leishmania* infections [21,22].

This study was designed to investigate how particular breeding sites may affect the biology of sand flies and their capacity to transmit *Leishmania* parasites, evaluating oviposition, life span, microbiome, and *Leishmania* infections in *L. longipalpis* reared on substrates collected from different ecological habitats in an endemic area of VL in North-Eastern Brazil.

## Methods

### Ethics statement

This work was conducted considering all ethical principles for animal experimentation and guidelines established by the Oswaldo Cruz Foundation (Fiocruz). All experiments involving mice were performed in accordance with the institutional review board (CEUA protocol 007/2016) of the Institute Goncalo Moniz (IGM–Fiocruz-Bahia/Brazil).

### Study sites

Collections were performed in Jauá and Jacuípe, coastal neighborhoods in Camaçari (12˚42'S and 38˚28'W), a city in the metropolitan region of Salvador, the capital of Bahia State-Brazil. The area is known to be endemic for VL in humans and dogs, with Camaçari having the highest number of VL cases [23,24]. Also, the coastal strip that includes the study area has been reported recently to be the largest concentration of the *L. longipalpis* population in the city [25].

### Insects

All experiments were performed using *L. longipalpis* reared in the insectary of the Instituto Gonçalo Moniz-FIOCRUZ-BA. The colony was under F9 generation and established using sand fly adults caught in different areas of Camaçari. Insects were reared at 70% relative humidity (RH), 25˚C, and a photoperiod of ≈12 h light/12 h darkness. Eggs harvested from oviposition pots were washed (2% sodium hypochlorite, 70% ethanol, and sterile water) to avoid contamination and sieved to remove the remains of dead adults. Larvae were fed on a mixture of rabbit chow and feces (1:1) prepared according to the method of Modi and Tesh (1983) with some modifications. The material was dried, mixed, crushed, sieved, and autoclaved before use. Adult sand flies were confined in fine mesh cages with free access to 30% autoclaved sucrose solution. Females were blood-fed on a Golden hamster anesthetized intraperitoneally with a combination of 150 mg Ketamine (Syntec, Brazil) and 10 mg of Xylazine (Syntec, Brazil) per kg.

### Substrates

Larval rearing substrates were collected from the surface to a depth of ~3cm of peridomestic habitats in Camaçari where adults of *L. longipalpis* were previously captured by Hoover Pugedo (HP) CDC-type light traps. The substrates included leaf litter (LL) of cashew trees, animal shelters (chickens, ducks), and tree holes. The material collected was air-dried, crushed, sieved inside a safety cabinet, and stored in 1.5 ml microtubes at -20˚C until use. The colony food described above was used as a reference in all experiments. The chemical and nutritional composition of each substrate was determined and parameters such as organic matter, pH, and protein content were analyzed in the Laboratory of Animal Nutrition, Federal University of Bahia, according to methods described by [26,27].

## Oviposition assays

Assays were performed in a three-choice test format comprising two different substrates and a control. Approximately 40 mm$^3$ of substrate was placed into a 50 ml polystyrene pot (Vero-copo, Brazil), the bottom of which was covered with ≈1 cm of heat sterilized (120˚C/2 h) plaster of Paris (SM Gesso, Brazil). A third small pot without larval substrate was used as a control. The plaster of Paris was moistened with approximately ≈2 ml of water before treatment. The pots were placed at equal distances inside an airtight clear plastic 500 ml container (Prafesta, Brazil) and covered with a fine mesh cloth with a hole to allow the release of insects. Three 4-day-old fed female *L. longipalpis* were aspirated into the plastic pot, the hole was plugged with cotton wool and the females were left inside for 72 h. Pots containing the pair-substrates tested were placed into a plastic box, then positioned inside an incubator in the dark under constant conditions of 25˚C and >70% RH. The total number of eggs inside each small pot was counted 72 h after the procedure, and each pair of substrates was tested in triplicate.

## Larval development on different substrates

Eggs randomly chosen from the colony were distributed into 500 ml polystyrene pots with the bottom covered with ≈ 2 cm of sterile plaster and observed every 12 h for emerging L1 larvae. As soon as larvae hatched from the eggs, groups containing one hundred L1 stage larvae were gently transferred to a new pot kept inside a plastic container. Larvae were fed *ad libitum* with one experimental substrate per pot, and three replicates under the same conditions were tested. Pots containing identical medium were grouped in the same plastic containers, and the external surfaces were disinfected continuously to prevent microbiota contamination. All materials were UV-sterilized before each test.

The development time of larval instars and the survival of adult sand flies were carefully tracked for two months after the emergence of the last adult to consider the individual variations of sand fly development. To evaluate adult survival, early emerged adults were individually transferred to 50 ml clear acrylic pots (MTEK, Brazil) covered with fine mesh cloth following the same standard insectary conditions as described previously. To observe possible differences in the adult survival attributable to their state of nutrition (larval diet), no sugar was offered during the experiment.

## Infection with *Leishmania infantum*

The efficiency of *L. infantum* infection was compared between *L. longipalpis* emerged from the chicken coop substrate and females fed with the colony food. Freshly isolated *L. infantum* amastigotes were cultured in Schneider's medium (Gibco BRL, New York, USA) supplemented with 20% fetal bovine serum (Gibco BRL, New York, USA) and 100 μg/mL gentamicin (Sigma Chemical Co., St. Louis, MO) at 26˚C. Female sand flies (4 days old) were fed with inactivated (56˚C for 1 h) rabbit blood infected with 10$^5$ promastigotes/mL through young chick-skin membrane stretched over custom-made glass feeders [28]. After 24 h, fully engorged females from both groups were transferred to new cages with free access to sterile 30% sucrose solution. Three- and nine-days post-infection females were dissected and observed using a phase-contrast microscope to evaluate the infection rate and colonization of the stomodeal valve. The parasite load per female was counted using the homogenate of each gut in 30 μl of 0.9% NaCl containing 2% of paraformaldehyde. The estimation of parasite number was performed on a hemocytometer considering any promastigote forms on day 3 and exclusively the metacyclic form on day 9. Some guts were also homogenized individually in 30 μl of sterile 0.9% NaCl solution to determine microbiota profiles by next-generation sequencing (NGS).

### Identification of microbiota

Larvae and adults reared on substrates were surface-sterilized (washed in 2% sodium hypochlorite, 70% ethanol, and sterile water) and their guts were carefully dissected under aseptic conditions. Guts were individually homogenized in 40 μl 0.9% NaCl saline and transferred to Whatman FTA Cards according to the manufacturer's instructions. DNA was extracted using phenol-chloroform [29]. All samples had the bacterial 16S rDNAv4 region amplified and larvae were pooled before library preparation for sequencing on the Illumina MiSeq platform using a paired-end protocol. Bioinformatic analysis was performed with Qiime version 1.9.1 [30], R's phyloseq [31] and microbiome [32] (Bioconductor, 2017–2019 microbiome R package. URL: http://microbiome.github.io). Barcodes and adapters had been previously removed from all forward and reverse fastq files which were joined and used for the OTU picking with the taxonomy-dependent closed-reference protocol. The sequences were clustered by uclust_ref using SILVA_V.123 bacterial reference alignment with an identity threshold for species-level assignment of 97%. Taxonomic α-diversity was measured within pools by the number of observed OTUs, Chao1, and Shannon indexes to measure species richness from each group. Weighted UniFrac β-diversity metric was assessed between samples using the β-diversity matrices to generate the Principal coordinates analysis (PCoA) plots for each comparison.

### Data analysis

All numerical data were tested for normality using the Shapiro-Wilk test. Kruskal Wallis test followed by Dunn's multiple comparison tests were employed to compare substrates to the cumulative number of eggs and the development of *L. longipalpi*s. The Spearman r test was used to evaluate the correlation between the number of eggs laid and the time required for insect development, as well as the correlation between substrate composition and insect development. Regarding the artificial infections with *L. infantum*, both groups were compared by the Mann-Whitney test. PCoA analysis was conducted to evaluate if any group of sand flies contained significantly different bacterial communities. The OTU abundance differences between sample pairs were assessed using bootstrapped Kruskal Wallis and p-value correction with FDR. The results are expressed as the group mean ± the standard error of the mean (SEM) and considered significant if $p < 0.05$, * denotes the significant difference. Statistical tests were performed using GraphPad Prism (version 8.0, USA).

## Results

### The effects of different substrates on development

In total, 1,717 eggs were laid by gravid *L. longipalpis*. The comparison of the number of eggs inside pots showed a significant correlation between the number of eggs and the type of substrate (Kruskal-Walls test, p <0.05). Fig 1A depicts oviposition data for the different tests, while relatively few eggs were oviposited in leaf litter (4.86%), a more substantial proportion was oviposited in pots containing colony food (33.06%) and material from chicken coops (36.93%) (Colony food, Chicken shelter vs leaf litter, Dunn's multiple comparisons test, p <0.05). The lowest number of eggs was oviposited inside pots containing only humid plaster of Paris (0.44%). These results confirm that females were attracted to pots containing oviposition substrates that are rich for larvae.

The time needed for larval development (stages L1 to L4) on different substrates was significantly different (p<0.05), while no effect was observed at the pupal (non-feeding) stage (p>0.05). Nonetheless, this was sufficient to affect the total duration of the insect lifecycle (p<0.05), with the first adults to eclose from pupated larvae reared on soil from chicken

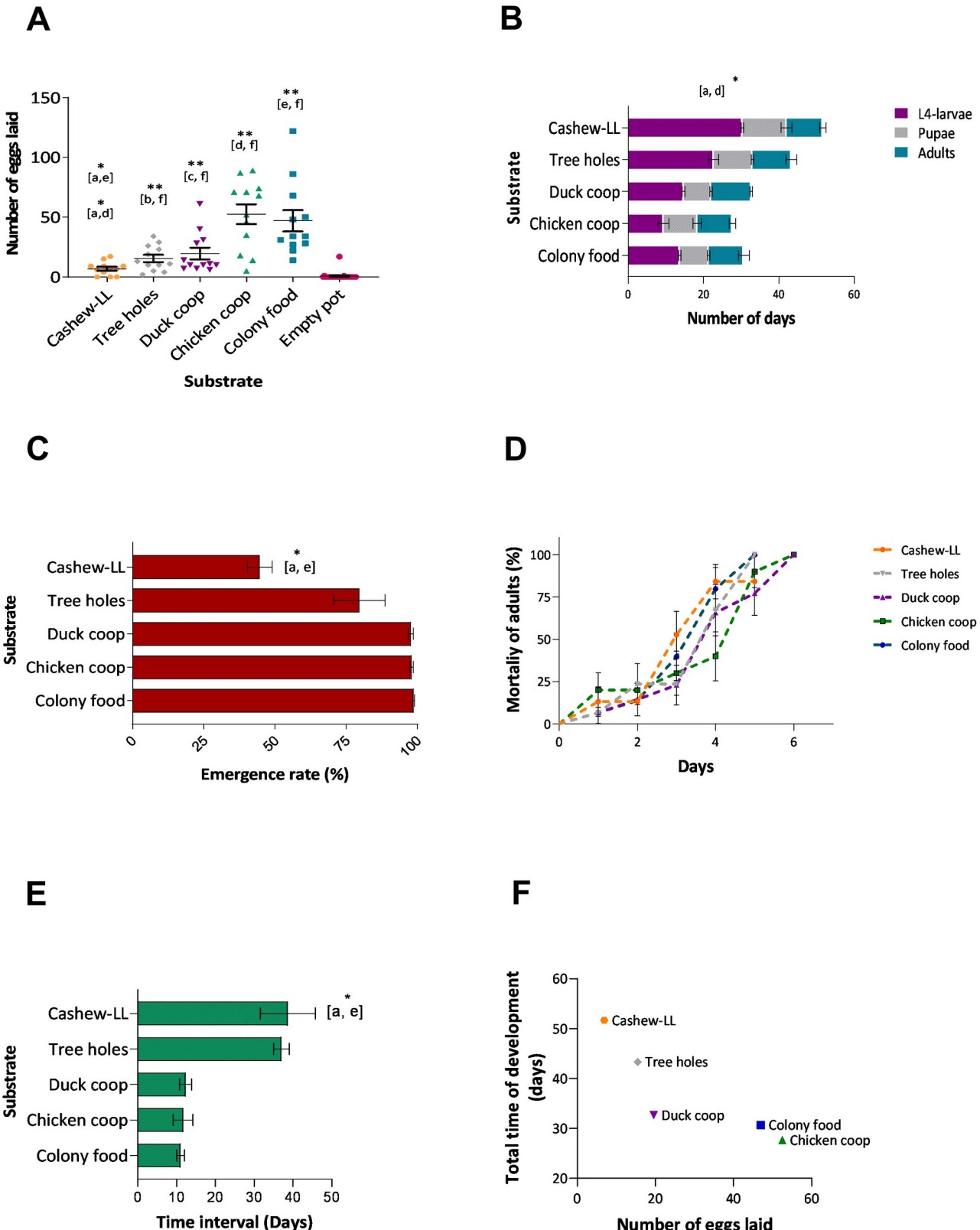

**Fig 1. *L. longipalpis* reared on different substrates collected from putative breeding sites in Camaçari.** A- Oviposition preference of colony reared flies on different substrates, B- rates of larval and pupal development on different substrates, C- proportions of eclosion of adults reared on different

substrates, D- survival of adults from each substrate, E- the time interval between eclosion of first and last adults, F- correlation between the number of eggs and total development time of insects on substrates. One hundred L1 larvae were transferred to plaster of Paris-lined pots and fed *ad libitum* with each type of substrate: a- cashew leaf litter, b- soil from inside tree holes, c- duck shelter, d- chicken shelter, e- colony food, and f- empty pot. Letters in brackets indicate the substrate-pairs with significant differences (Dunn's multiple comparisons test). Results are expressed as the group mean ± (SEM) * P<0.05, **P <0.01.

shelters within 28 days, whereas it took twice as long (50.3 days) for sand flies grown on LL (Fig 1B). Also, sand flies reared on leaf litter exhibited the longest interval between the first and last adults to eclose from pupae (p<0.05), with a significantly lower proportion of the larvae reaching maturity (p<0.05) (Fig 1C and 1E). Only 44.6% of L1 larvae reared on leaf litter reached the adult stage, whereas 98.6% of larvae reared on colony food emerged as adults (Dunn's test, Colony food vs leaf litter, p>0.05). Despite these differences among groups, the survival of adults was not affected by the type of larval substrate used (Log-rank test for trend, p>0.05) (Fig 1D).

Interestingly, there was an inverse correlation between the substrates when the number of eggs was compared to the total time of insect development (Spearman r test, r = -1, p<0.05) (Fig 1F), the greater the number of eggs oviposited by females, the shorter the time for insects to reach adulthood, as well as larval stage L4 (Spearman r test, r = -1, p<0.05).

To test if the extraordinarily long development of larvae reared on cashew leaf litter was characteristic of this plant species, the development of *L. longipalpis* larvae on leaf litter collected from under other trees common to the endemic area, *Mangifera indica* (mango tree) and *Citrus sinensis* (orange tree), was assessed, confirming that larvae development was affected by the tree species leaf litter (p < 0.05) (Fig 2A). Larval development was significantly faster on orange and mango than on cashew leaf litter (Dunn's multiple comparisons test, cashew tree vs orange tree, p < 0.05). To assess whether factors other than tree species could affect larval development, *L. longipalpis* larvae were reared on similar substrates collected in different locations but there were no significant differences between sites (Fig 2B).

## Physicochemical analysis of substrates

The physicochemical analyses revealed that most substrates were slightly acidic but some were neutral (pH 5.6–7.5), with significant variations in the proportion of organic matter and proteins (Table 1), however, there was no correlation between any of the parameters measured and the time required for larval development (p>0.05), pupae (p>0.05), or adult (p>0.05).

## *Leishmania infantum* infection

Experimental infections with *L. infantum* were performed using females emerging from the chicken shelter experimental group and compared with those reared under colony conditions, showing no significant differences between groups (Fig 3). On day 3 post-infection, promastigote forms of *L.infantum* were observed in the anterior midgut of approximately 80% of the flies from both groups. Nine days post-infection, flies from the chicken coop substrate harbored a larger number of metacyclic forms on average than those reared on colony larval food but this did not reach statistical significance. Similar numbers of metacyclic promastigotes were observed in the stomodeal valve of females in both groups (p>0.05).

## Microbiota analysis

The gut microbiota of *L. longipalpis* larvae differed between substrates, with differences also observed between L4 larvae and adults from the same experimental group (Fig 4). The

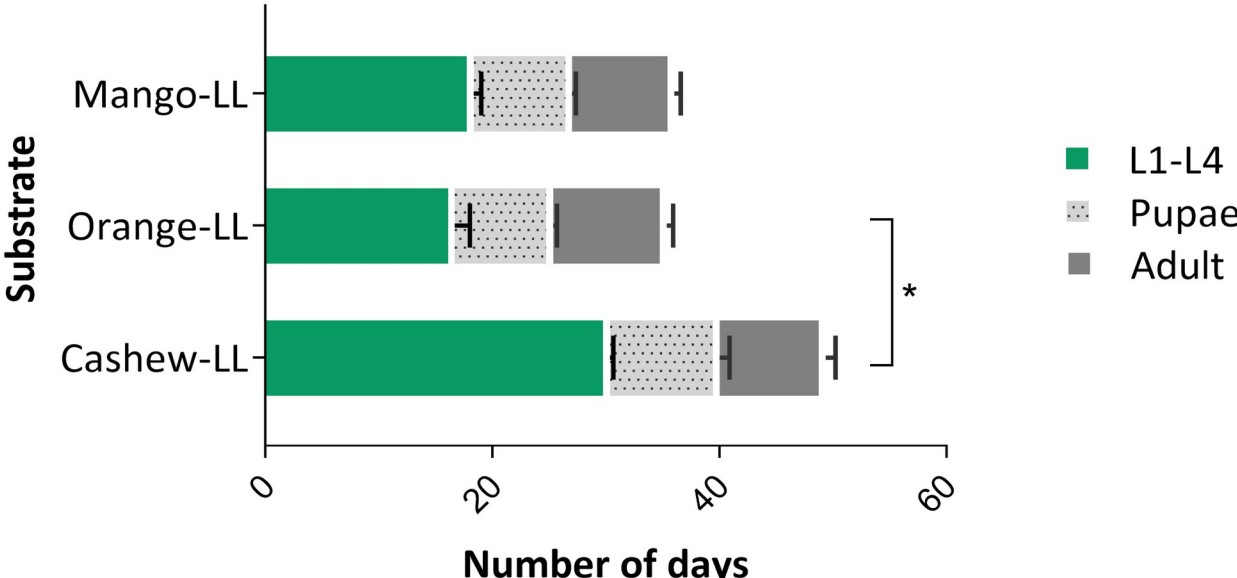

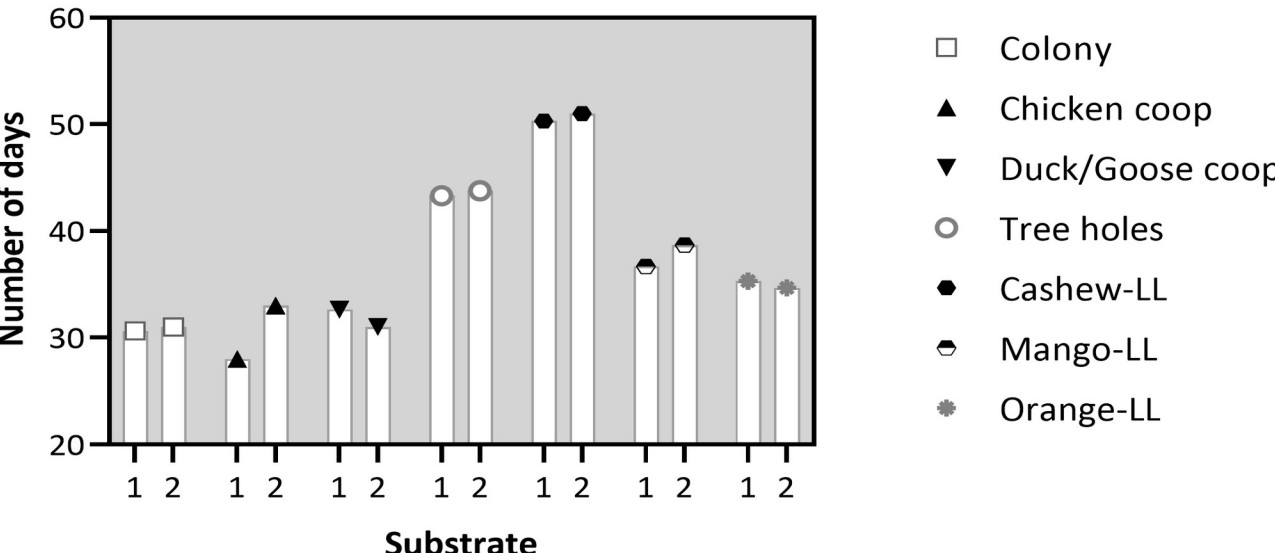

**Fig 2. Development of *L. longipalpis* on different substrates collected in the endemic area of Camaçari.** A- Development of *L. longipalpis* in substrates of different leaf litter from different species of tree, B- comparison of the development of *L. longipalpis* in substrates collected in different peri domiciles. The data represent the time interval of development of insects from the L1-larvae recently hatched until the adult stage of *L. longipalpis*.

**Table 1. Physiochemical parameters of the larval feeding substrates.**

| Substrate | pH | Dry weight (%) | Protein (%) | Organic matter (%) |
|---|---|---|---|---|
| Colony food | 6.7 | 89.77 | 12.37 | 89.92 |
| Chicken shelter | 7.5 | 93.12 | 2.555 | 9.435 |
| Duck/Goose shelter | 6.8 | 83.24 | 1.97 | 5.27 |
| Tree holes | 6.8 | 99.76 | 1.98 | 1.8 |
| Cashew-LL | 6 | 96.16 | 3.12 | 80.08 |
| Mango-LL | 5.6 | 95.02 | 4.49 | 77.28 |
| Orange-LL | 7.8 | 97.1 | 4.16 | 32.82 |

bacterial genus *Cupriavidus* was the most abundant in all groups, with the highest proportion in larvae reared on chicken coop substrate. *Methylobacterium* was present in the highest proportions in adult and L4 larvae reared on cashew leaf litter. *Wolbachia* was present in all groups with higher abundance in larvae reared on orange leaf litter and adult sand flies reared on chicken coop substrate.

The comparison of midgut microbiota of sand flies regardless of lifecycle stage showed that *Escherichia-Shigella*, *Methyloversatilis*, *Massilia*, *Ralstonia*, *Cupriavidus*, *Methylobacterium*, *Ochrobactrum*, *Fusobacterium*, and *Hydrotalea* were more common in sand flies reared on cashew leaf litter, whereas *Lawsonella* and *Staphylococcus* were most abundant in sand flies reared on chicken coop substrate and orange tree leaf litter, respectively. However, these differences did not reach statistical significance (Fig 5).

The beta diversity's PCoA analysis of the microbiome of L4 larvae showed specific profiles typical of the substrate upon which they had been raised (Fig 6A). On the other hand, the same analysis performed on adults showed that the microbiome of adult sand flies reared on cashew leaf litter was significantly different from the microbiomes of sand flies reared on colony food, mango and orange leaf litter (Fig 6B). The microbiomes of adults reared on these three substrates were closely related, congregating in the same region of the PCoA plot.

Regarding the most abundant bacterial genera in guts of larvae and adult *L. longipalpis* reared on different substrates (Fig 6C–6F), there were no differences in the abundance of

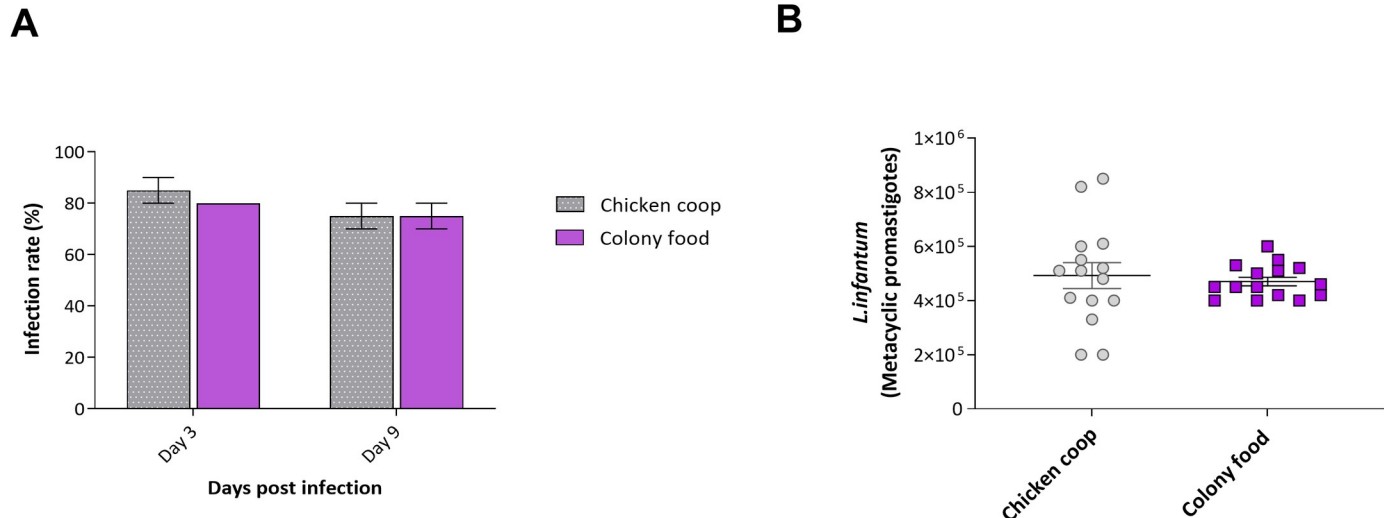

**Fig 3. Infection of female *L. longipalpis* reared on chicken house substrate or colony food with *L. infantum*.** A- the proportion of infected females 3- and 9-days post-infection, B- Parasite load in females infected 9 days post-infection. Bars represent the group mean ± SEM, n = 2.

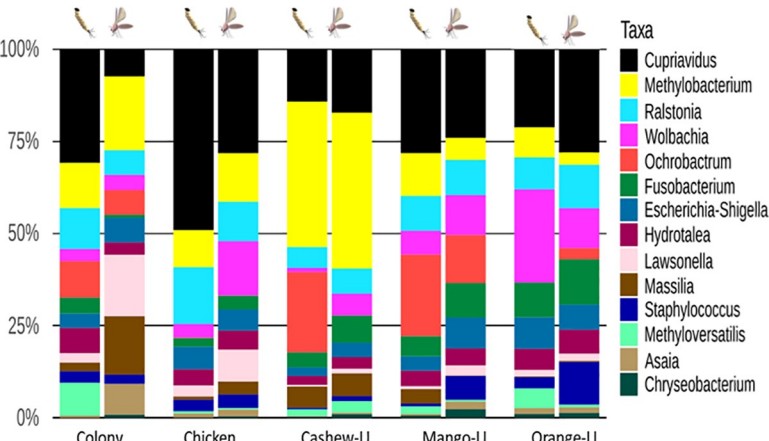

**Fig 4. Relative abundance of bacteria found in the midguts of L4 larvae and adult *L.longipalpis*.** The bar graph depicts the relative abundance of 16S OTUs in sand flies reared on different substrates. Bacterial genera are listed according to their abundance in each group.

*Cupriavidus* and *Ralstonia* but the proportion of *Wolbachia* was statistically lower in L4 larvae reared on chicken coop substrate and adults emerging from the mango leaf litter group (p<0,05) (Fig 6E). The genus *Methylobacterium* was the most abundant in both larvae and adults reared on cashew leaves, being statistically higher in comparison with sand flies reared on chicken coop substrate both for L4 and adult midguts (Fig 6D). *Methylobacterium* abundance was lower in adults reared on mango and orange leaves when compared with larvae reared on cashew leaf litter. Similarly, the abundance was lower in flies reared on orange-tree leaf litter and adults reared on mango leaves in comparison with adults reared on cashew substrate.

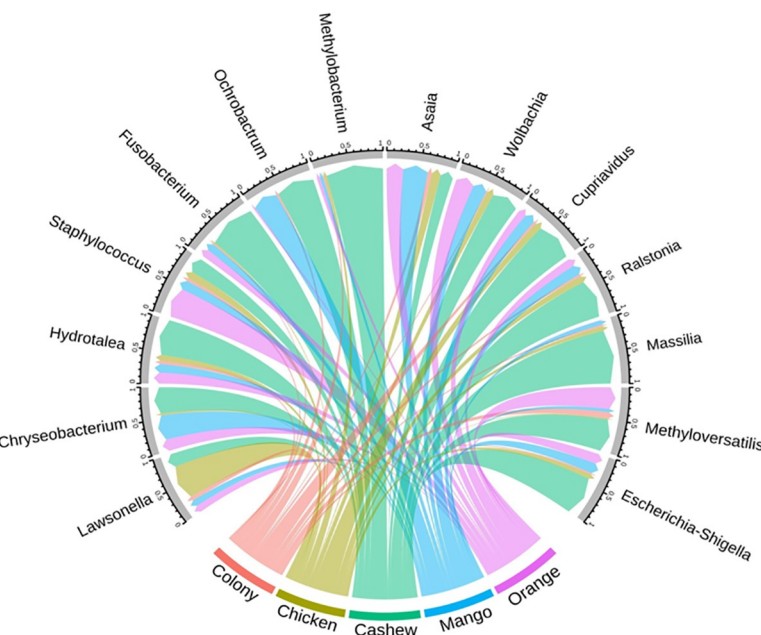

**Fig 5. Circular plot depicting the relative abundance of different bacterial genera in adult sand flies (gray border) reared as larvae on different substrates.**

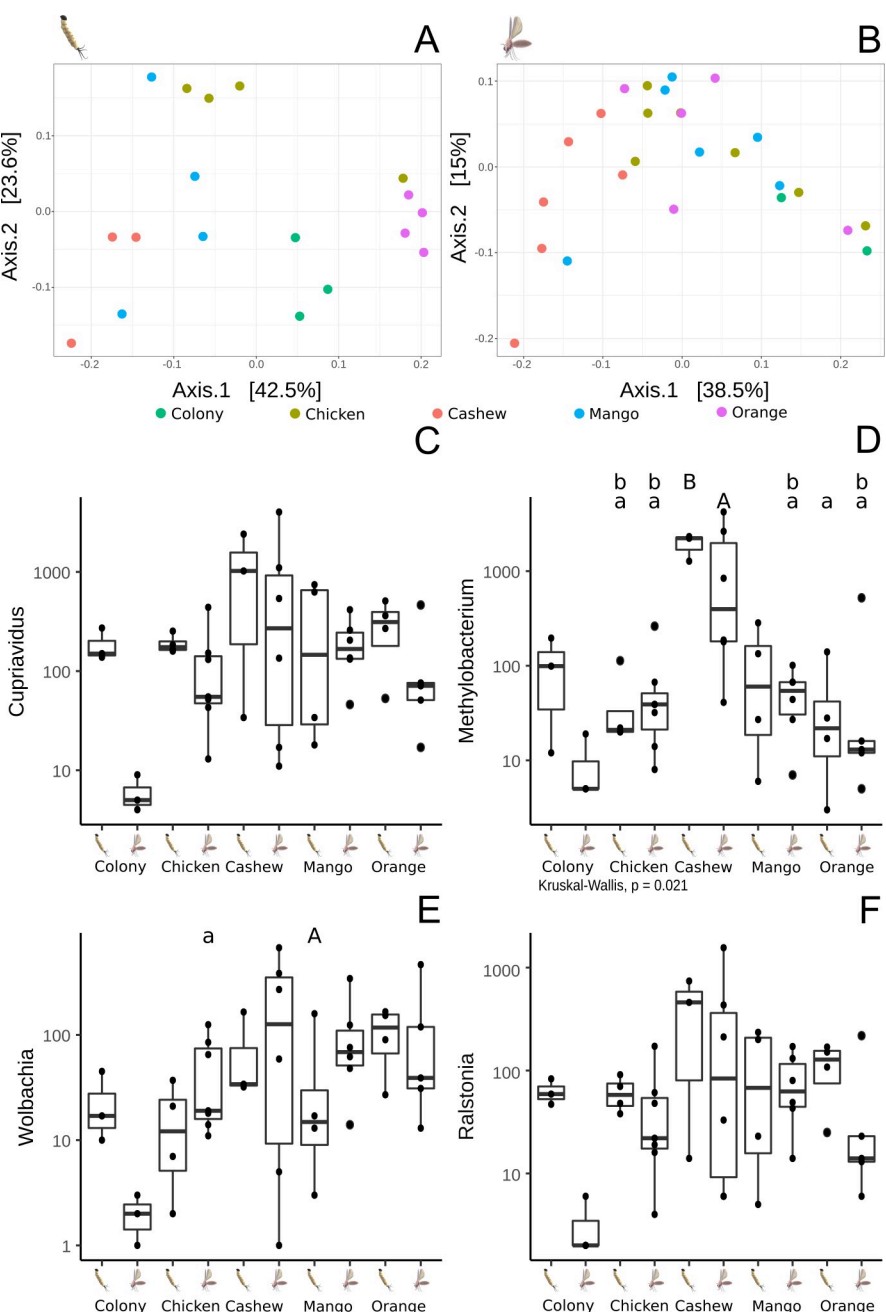

**Fig 6. Dominant bacterial genera found in the guts of *L. longipalpis*: comparison of different lifecycle stages reared upon different substrates.** A- PCoA depicting the β diversity (between groups) of midgut microbiota of L4 larvae reared on different substrates. B- PCoA β analysis of the midgut microbiota of adults derived from larvae reared on different substrates. C- Presence of *Cupriavidus* in larvae and adults reared on the different substrates. D- Boxplot representing the presence of *Methylobacterium* in larvae and adults. E- Proportion of *Wolbachia* in larvae and adults reared on the different substrates. F- Presence of *Ralstonia* in larvae and adults reared on the different substrates. Letters above boxplots represent statistically significant differences (Wilcoxon, p<0.05) in a pairwise comparison, i.e. lowercase letters statistically less frequent than uppercase.

With regards to the bacterial richness in midguts of adults reared on different substrates, the diet of orange leaves resulted in enriched microbiota in L4 larval midguts compared to mango leaf and chicken substrate (Fig 7). Such differences were also observed in midguts of

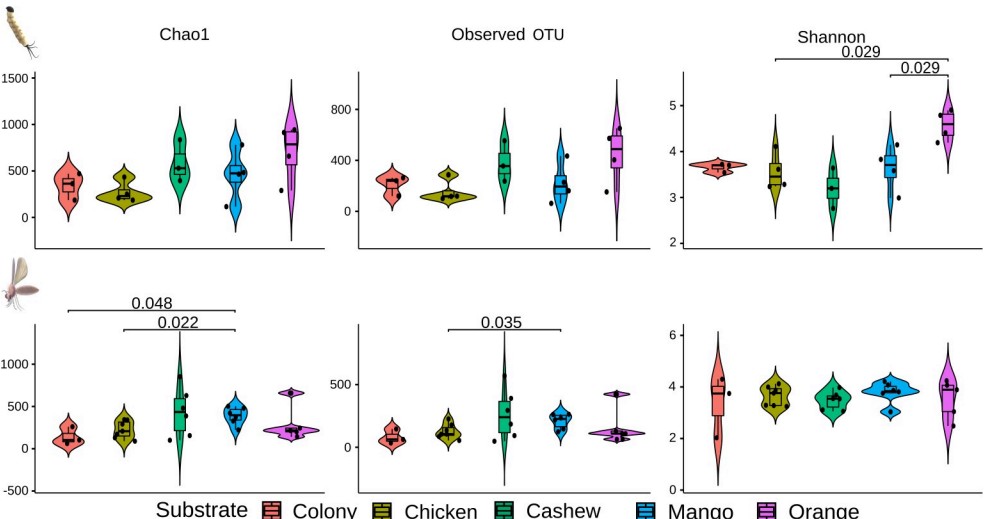

**Fig 7. Violin boxplot of the alpha diversity of bacterial composition in midguts of L4 larvae and adults reared on different substrates.** Species richness was assessed by Chao1 (estimates diversity from abundance data), Observed OTU (sum of unique OTUs in each sample), and Shannon-Wiener (sum of species proportion logarithms) indexes. Higher index values indicate richer and more diverse bacterial composition.

adults reared on mango-leaves, which induced a richer microbiota when compared with the colony and chicken substrates using the Chao1 and Observed OTU metrics.

## Changes in the microbiota of sand flies infected with *L. infantum*

Analysis of the bacterial microbiota in sand flies guts before and during infection with *L. infantum* showed that in the colony flies the genera *Massilia* and *Methylobacterium* were predominant in unfed adults, but declined significantly by the third day after infection with *L. infantum* (Fig 8A). While *Methylobacterium* resurged on the 9th day post-infection, the abundance of *Massilia* remained low. *Methyloversatilis* showed higher abundance on the latest evaluated time point. The microbiota of unfed and infected blood-fed sand flies grown on the chicken substrate was similar as illustrated by the PCoA analysis depicted in (Fig 8B). Microbiota of flies reared on chicken-coop substrate congregate closely. In contrast, the microbiota of the colony group was less uniform and spread out (Fig 8B). Interestingly, the adult midgut bacterial microbiota on the 3rd-day post-infection was similar in both groups of infected flies, with no statistically significant difference in the microbiota of unfed, 3rd, and 9th-day post-infection adults reared on either colony or chicken coop substrates.

## Discussion

### Performance of insects

This study investigated the influence of larval diets on the biology of larval *L. longipalpis*. The vast number of eggs oviposited on the chicken coop substrate confirms that *L. longipalpis* females preferentially select oviposition sites that offer optimal conditions for the developing larvae, probably via chemosensory olfactory and visual cues [33]. The experiments were conducted in complete darkness, indicating that visual stimuli were probably not a relevant clue for their behavior (Fig 1A).

 The females preferred pots containing substrates from colony food (rabbit chow/rabbit feces 1:1) and chicken shelters, which is consistent with previous work demonstrating that

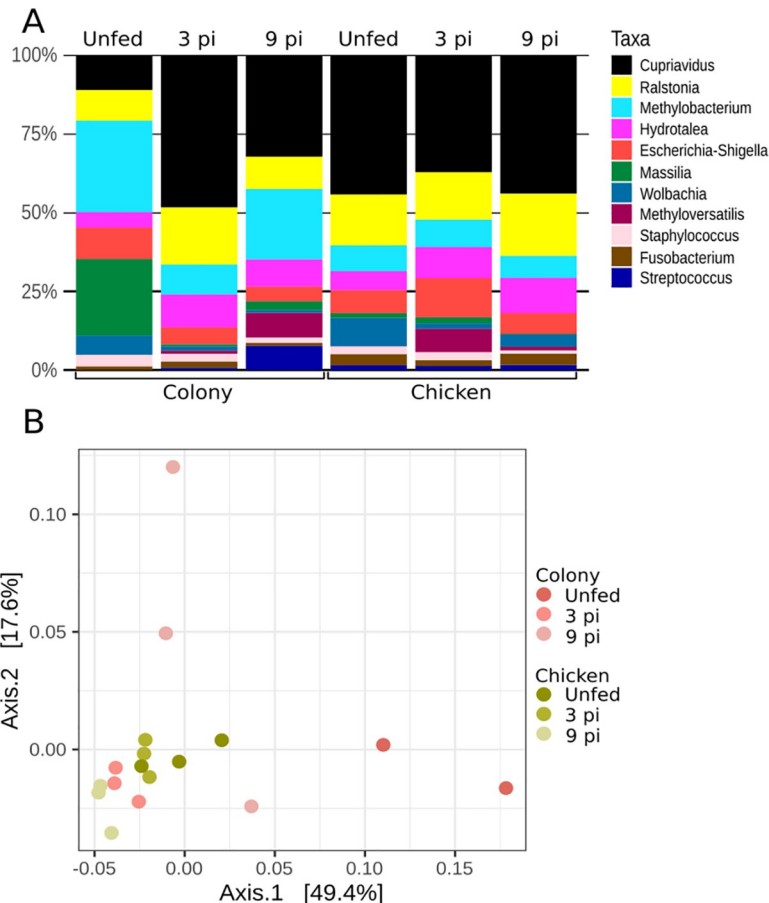

**Fig 8. Midgut microbiota of *L. longipalpis* females reared on different substrates before and during infection with *L. infantum*.** A- Relative abundance of bacterial genera and B- PCoA of βdiversity of bacterial genera (3pi and 9pi = 3rd and 9th day post-infection, respectively).

receptors of gravid *L. longipalpis* females were stimulated by hexanal and 2-methyl-2-butanol present in chicken and rabbit feces [34]. The microbial communities on the larval substrates can also play an essential role in the inducement of oviposition by sand fly females [35]. However, there was no correlation between the gut microbiota of sand flies with chicken shelter or colony substrates.

Predictably, only larval stages were affected by the substrate upon which they were reared, but it was sufficiently profound to significantly affect the duration of the entire life cycle (Fig 1B). Moreover, the observed time lag between the first and last larvae to pupate as well as the low hatching rate on cashew leaf litter, indicates that larval feeding on substrates that are less than optimal would be deleterious to the adult population of *L. longipalpis*.

The most evident difference was observed between sand flies reared on chicken coop substrate and those reared on cashew leaf litter. The adult *L. longipalpis* eclosing from larvae reared on the chicken coop substrate were robust and presented the usual dark brown color, whereas those reared on cashew leaf litter were lighter colored with smaller bodies (S1 Fig). The highest number of eggs were oviposited, and larvae developed faster in the chicken coop substrate. In contrast, gravid females confined to pots containing leaf litter from cashew trees oviposited fewer eggs and the larval development period was significantly prolonged (Fig 1A and 1B). This inverse correlation supports the hypothesis that female *L. longipalpis* select

oviposition sites that contain optimal larval food substrates. Selective oviposition has been described in other insects and previously suggested in sand flies [35,36,37,38].

Selective oviposition in optimal ecotopes is probably essential for maintaining the population of *L. longipalpis* in environments such as the study area, a coastal region with predominantly sandy landscapes. In such soils, there are fewer nutrients, lower moisture retention and they are deficient in organic matter. Indeed in a recent study in the same area, *L. longipalpis* populations were denser in the peridomestic habitats of the coastal areas of Camaçari while fewer flies were found in the pristine habitats of the same region [25].

### Physicochemical analyses of larval growth media

There was a large variation in the physiochemical parameters of the substrates tested, however, none of the parameters correlated with optimal *L. longipalpis* larval development (Fig 1B and Table 1). Previous studies suggested that although sand fly larvae need organic matter for their development, high levels are not necessarily required to attain adulthood [18,39]. Given the fact that all substrates used were collected in putative larval breeding sites, we expected to find satisfactory conditions for development in all of them. For example, despite the adequate nutritional value of the cashew leaf litter substrate, larval development was suboptimal (Fig 1).

Previous studies have described the presence of sand flies, including *L. longipalpis*, in habitats with decaying leaves [17,40], so we investigated whether leaf litter from different tree species had different effects on the larval development of *L. longipalpis*. While larval development on mango and orange leaf litter was relatively normal, cashew leaves were suboptimal. The cashew nut tree is native to Northeast Brazil and commonly found in the vicinity of houses in the study site. Cashew trees are toxic to some insect species, with a profound deleterious effect on mosquito larvae [41] due to the high concentration of tannins, oxalate, stearic acid, glucuronic and glutamic acids in cashew leaves. Moreover, the oviposition and progeny development of the beetle *Callosobruchus subinnotatus* is severely suppressed by components found in cashew plants [42]. Hence, despite having sufficient nutritional value for the development of sand fly larvae, cashew leaf litter may contain toxic substances that hinder optimal larval development. The potential of components from this plant against sand flies is an aspect that deserves to be evaluated as a possible biological control.

### Microbiota community

Distinct microbiota profiles were characterized in larvae and adult *L. longipalpis* sand flies (Fig 7). Although trans-stadial bacterial colonization of the adult gut is known to occur in sand flies, most microorganisms are eliminated due to physiological changes during metamorphosis [20,43]. Despite the significant variations in microbiota among L4 larvae reared on different substrates, all experimental groups shared a common core of microorganisms, including *Cupriavidus*, *Ralstonia*, *Wolbachia*, and *Methylobactetium* spp (Fig 7). Most of these genera were previously described in wild-caught *Lutzomyia* sp. sand flies, except *Cupriavidus* [44,45]. Surprisingly, in our study, *Cupriavidus*, known for extreme diversity in suitable habitats such as soil and plants [46,47], was found in a large proportion in all experimental groups (Fig 4). Although larvae reared on cashew leaf litter were frequently infected with all the dominant bacterial genera, the genus *Methylobacterium* was only found in a significant proportion of larvae and adults in this group (Fig 6D). *Methylobacterium* spp. are commonly isolated from various natural environments, including plant leaf surfaces and soil [48], and has been identified in wild-caught as well as lab-reared sand flies [21,49]. Our results showing the presence of *Methylobacterium* in females from colony and field substrate after the infection with *L. infantum* support the suggestion of [50], that *Methylobacteriaceae* remains in the sand fly midgut in

the presence of *Leishmania* infections. *Methylobacteriacae* were exceptionally extant in poorly developing larvae in sand flies reared on cashew leaf litter, so it would be worthwhile to explore whether the harmful effect is attributable to *Methylobacterium* or innate chemical properties of cashew leaves.

The lack of a direct comparison between the microbiota found in substrates collected and sand flies was a limitation in this study. This analysis was prevented by poor quality substrate samples. Nonetheless, it is known that the diet of sand fly larvae and adults play a major role in their microbiome community [51] and all L1-larvae were hatched from disinfected eggs and developed on sterilized material, as well as having the same origin, age, and equal treatments. The source of larval food from colony or collected from field environments was the only distinctive aspect. Considering such context, we believe that significant differences observed in the larvae beta diversity's PCoA analysis from larvae (Fig 6A) are strongly related to each substrate offered. This difference in microbiota diversity reduced when insects became adults but was expected due to physiological changes during the pupae phase.

The female *L. longipalpis* reared on colony or chicken coop larval substrates were highly susceptible to infection with *L.infantum*. Moreover, the microbial profile in both groups was similar, suggesting that bacterial genera have no deleterious influence on the susceptibility of *L. longipalpis* for *L. infantum*. The comparison of groups prioritized only highly abundant bacteria and those genera are found broadly in sand flies, however, possible lab contamination during sampling should also be considered.

## Infections with *L. infantum* and epidemiological meaning

The development of *L.infantum* infections in females grown on colony food and chicken coop substrates was very similar, with high infection rates and parasite loads, and the appearance of infective metacyclic promastigotes coincided (**S2 Fig**). Although we did not conduct sand fly transmission experiments, our results suggest that females reared on these two substrates would transmit *L.infantum* efficiently. Unfortunately, due to low numbers of emerging adult females, it was not possible to study parasite development in flies reared on cashew leaf litter, hence whether suboptimal larval rearing conditions affect the adult females' vector potential.

The apparent suitability of chicken coops for larval breeding was reflected by the large number of adult sand flies captured by light traps in and near coops compared with other environments from the same backyard. As it was not possible to determine whether these adults emerged in the chicken coop or were attracted to the chickens as a source of blood, this information was excluded from comparisons. The presence of *L. longipalpis* in chicken coops is usually attributed to their attraction to chickens for blood feeding [52]. Chickens are diurnal and sleep at night when sand fly females seek blood. Moreover, chicken combes provide ample exposed skin for biting, making these birds especially suitable blood hosts for biting sand fly females [53]. Our results show that the presence of sand fly adults in chicken coops may also reflect suitability for larval breeding. These hypotheses were tested and confirmed by [54] who used emergence traps to demonstrate high numbers of sand flies emerging in chicken coops.

The optimal development of *L. longipalpis* larvae reared on chicken coop substrates reaffirms the role of chickens in endemic areas of VL [52,53,55,56]. Chicken coops are very common and numerous in many rural and suburban areas. The high attractiveness of oviposition and efficient development in chicken coop substrates supports the hypothesis that such sites are a great breeding site for *L.longipalpis*. According to our results, this environment would promote a larger population of sand flies in a shorter period compared to the other explored habitats. It should be meaningful to maintenance of sand flies densities on endemic areas, and focused actions in such environments could be an efficient strategy for vector control.

The susceptibility of females reared in chicken coops for *L. infantum* infection also supports the epidemiological importance of this habitat to the transmission of *L. infantum* to dogs and humans. In theory, sand fly females hatching in chicken coops may feed primarily on these birds, and since birds do not become infected with any *Leishmania* species, chickens may contribute to zoo-prophylaxis. At the same time, sand fly females only feed on blood 3–4 days post-emergence [57] and as they require carbohydrates, they may forage away from the chicken coop seeking sugar meals, thus, dispersal in search of carbohydrate sources could increase their chances of feeding on hosts other than chickens.

This study showed for the first time that the source of food present in putative breeding sites of the sand fly *L. longipapis* can have a distinctive effect on its life cycle, suggesting a direct influence of breeding site type on the population density of *L. longipalpis* in endemic areas, and possibly, in the epidemiology of the disease. Considering that selection of females for oviposition is linked to the success of further insect life stages, breeding sites used by sand flies in nature are likely less random and diverse than first believed. We suggest that further studies evaluating the efficiency of different breeding sites can offer the opportunity to focus the actions of entomological control for this vector in urban areas.

## Supporting information

**S1 Fig. Appearance of adult *L. longipalpis* reared in different substrates.** Males and females reared on substrates from A) chicken coops (left), and B) composting Cashew tree leaf litter (right).
(TIF)

**S2 Fig. Images of the stomodeal valve of *L. longipalpis* on 9th day post-infection with *L. infantum*.** A) dissection of female reared in chicken coops, and B) dissection of female reared with colony food.
(TIF)

**S1 Data. Spreadsheet of data analyzed and illustrated.** Each separate sheet details figures data of the oviposition assays, the development of *L. longipalpis* on substrates, infections of *L. infantum* and microbiota analysis.
(XLSX)

## Acknowledgments

The authors wish to thank the Laboratório de Nutrição Animal/Universidade Federal da Bahia-UFBA, for the physicochemical analyses of substrates.

## Author Contributions

**Conceptualization:** Kelsilandia Aguiar Martins, Alon Warburg.

**Data curation:** Kelsilandia Aguiar Martins, Maria Helena de Athayde Meirelles, Tiago Feitosa Mota, Ibrahim Abbasi.

**Formal analysis:** Kelsilandia Aguiar Martins, Maria Helena de Athayde Meirelles, Tiago Feitosa Mota, Ibrahim Abbasi, Artur Trancoso Lopo de Queiroz.

**Funding acquisition:** Claudia Ida Brodskyn, Patrícia Sampaio Tavares Veras, Alon Warburg.

**Investigation:** Kelsilandia Aguiar Martins, Maria Helena de Athayde Meirelles.

**Methodology:** Kelsilandia Aguiar Martins.

**Project administration:** Patrícia Sampaio Tavares Veras, Deborah Bittencourt Mothé Fraga.

**Supervision:** Artur Trancoso Lopo de Queiroz, Deborah Bittencourt Mothé Fraga, Alon Warburg.

**Writing – original draft:** Kelsilandia Aguiar Martins, Tiago Feitosa Mota.

**Writing – review & editing:** Claudia Ida Brodskyn, Patrícia Sampaio Tavares Veras, Deborah Bittencourt Mothé Fraga, Alon Warburg.

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
