## [Decision Letter · Decision Letter 0]

8 Sep 2020

Dear Dr Aguiar Martins,

Thank you very much for submitting your manuscript "Effects of larval rearing substrates on some life-table parameters of Lutzomyia longipalpis sand flies" for consideration at PLOS Neglected Tropical Diseases. As with all papers reviewed by the journal, your manuscript was reviewed by members of the editorial board and by several independent reviewers. In light of the reviews (below this email and in a separate file), we are likely to accept your manuscript provided that a significantly-revised version is submitted that takes into account the reviewers' comments. 

The reviewers raised several constructive comments that will be very useful for improvement of the manuscript, including providing more background, detail to the methods section, results (e.g. location of infection, proportion of metacyclics) and more elaborate discussion of the results. Please find the comments of referee 2 as a separate file.

We cannot make any decision about publication until we have seen the revised manuscript and your response to the reviewers' comments. Your revised manuscript is also likely to be sent to reviewers for further evaluation.

Sincerely,

Guy Caljon

Associate Editor

Abhay Satoskar

Deputy Editor

Editorial comment:

In addition to the reviewers' comments, I would suggest to modify the graphs in Fig. 1A, 3A, 6C-F, 7 to overlay the graphical representations with the individual data points.

Reviewer's Responses to Questions

Key Review Criteria Required for Acceptance?

Methods

-Are the objectives of the study clearly articulated with a clear testable hypothesis stated?

-Is the study design appropriate to address the stated objectives?

-Is the population clearly described and appropriate for the hypothesis being tested?

-Is the sample size sufficient to ensure adequate power to address the hypothesis being tested?

-Were correct statistical analysis used to support conclusions?

-Are there concerns about ethical or regulatory requirements being met?

Reviewer #1: The study is based on sufficient data amount and the study design is appropriate, but the following issues should be completed/added: 

Page 8: the description of the method of determination of metacyclic stages in sand flies is missing. 

Identification of the microbiota was done from homogenized guts of larvae and adults reared on different substrates. However, the microbiota pattern in the field substrates was not analysed. In my opinion, this information is substantial and should be provided as it allows comparison of the relative representation of the bacterial genera in the food and sand fly guts (particularly interesting for the representation of Methylobacterium sp. discussed on Page 22)

Reviewer #2: Yes, to all questions.

Reviewer #3: (No Response)

Results

-Does the analysis presented match the analysis plan?

-Are the results clearly and completely presented?

-Are the figures (Tables, Images) of sufficient quality for clarity?

Reviewer #1: The results are clearly presented and the figures are of sufficient quality. However, some details on Leishmania infection are missing:

Page 13, lines 281-290 and Fig. 3. There is no information on the colonization of the stomodeal valve, which is the prerequisite for the successful transmission to the host. Could authors complete the location of infections either in the form of the figure or just in the text? Presence of metacyclic forms is not specified sufficiently - please provide the relative representation of these forms in both groups.

Reviewer #2: Yes, to the first two questions.

No, to the third question

Reviewer #3: - The legend in Figure 3 is swapped: A) represents the parasite load and B) represents the proportion of infected females

- There is no legend for S1 Figure1

Conclusions

-Are the conclusions supported by the data presented?

-Are the limitations of analysis clearly described?

-Do the authors discuss how these data can be helpful to advance our understanding of the topic under study?

-Is public health relevance addressed?

Reviewer #1: See the Summary and General Comments section

Reviewer #2: Yes, to all questions.

Reviewer #3: In the introduction the authors mention that sand fly breeding sites could be an alternative target for control of leishmaniasis, however in the discussion I expected that the authors would discuss this topic further. Since this research demonstrates that chicken-coop substrate contribute to the development of L. longipalpis and probably to the transmission of L. infantum to dogs and humans, I would suggest to add a few lines about control of these breeding sites in the elimination of Leishmania.

**Editorial and Data Presentation Modifications?**

**Use this section for editorial suggestions as well as relatively minor modifications of existing data that would enhance clarity. If the only modifications needed are minor and/or editorial, you may wish to recommend “Minor Revision” or “Accept”. **

**Reviewer #1: Abstract: As no significant differences were found between the experimental group and the control group in susceptibility to Leishmania, the expression about high susceptibility of the experimental group (line 36) is misleading and should be changed. Better to use the same wording as in the Results (page 13) or Discussion (page 20). **

**The terms “leishmaniasis” and “visceral leishmaniasis” should be written consistently with lower-case letters throughout the text (please correct on lines 27, 31, 45, 49, 109-110)**

**Page 4, line 75: in addition to the reference No. 7, also a more recent publication(s) on the development of L. infantum in L. longipalpis and the transmission by the bite should be added (Secundino et al. 2012; Parasites & Vectors 2012, 5:20, Freitas et al. 2012, Am J Trop Med Hyg 2012, 86:4)**

**Page 4, line 88: Consider replacing the reference No. 14 with another one, where life span is given for various sand fly species, including L. longipalpis (Volf and Volfova, 2011. J Vector Ecol). Then, you may also discuss if developmental times in your experiments are different from those described in this paper.**

**Page 5, lines 98-99: the references 21 and 22 do not contain single evidence about the effect of microbiota on sand fly susceptibility to Leishmania infection. Please correct the sentence or use other references like [53] Kelly et al. 2017, Sant’Anna et al. (2014) Parasites & Vectors 7:329.**

**Page 8. Line 172 – the reference No. 27 does not relate to the method of sand fly infections - should be replaced with a more relevant one.**

**Page 16 line 373 – I suppose the word “larval” is a typing error as the whole chapter refers to the adult microbiome. **

**In the References section, there are many errors in the use of italics and capitals which must be corrected.**

**Reviewer #2: N/A**

**Reviewer #3: - line 160 "... were carefully monitored for two months after the last adult had from the pupa." There is a missing verb in this sentence.**

**- line 499 "In females of L. longipalpis...." replace by "Females of L. longipalpis..."**

**--------------------**

**Summary and General Comments**

**Use this section to provide overall comments, discuss strengths/weaknesses of the study, novelty, significance, general execution and scholarship. You may also include additional comments for the author, including concerns about dual publication, research ethics, or publication ethics. If requesting major revision, please articulate the new experiments that are needed.**

**Reviewer #1: The study evaluates the influence of the larval food, associated with different breeding sites, on the development, microbiome and susceptibility of L. longipalpis to Leishmania. Results of the oviposition assay confirm a well-known fact that sand fly females prefer breeding sites with organic matter rich in nutrition for larvae. Substrates used for larval rearing differed in many parameters; therefore, it is not surprising that these food types influenced the length of the larval development. On the other hand, the life span of pupae and adults did not differ significantly. The comparison of susceptibility to Leishmania did not reveal any differences between the control group and the group reared on the soil from the chicken shelter. This result is also expectable, as larvae reared on this breeding material developed best from all the tested groups. Evaluation of the susceptibility of females reared on the poor substrate represented here by the cashew litter would be more interesting, but, as written in the Discussion, the numbers of females emerging on this material were too low to allow the experiment. The analysis of the microbiome showed distinct profiles in both larval stages and adults. However, the results might be better understandable and discussible if authors also provided the analysis of microbiota occurring in the substrates offered to larvae. **

**Generally, the manuscript is well written but several methodological issues needs to be completed before accepting for publication: identification of microbiota in field substrates, location of Leishmania infection and representation of metacyclic forms in the sand fly midgut. The novelty and significance of findings are limited and data are interesting for a limited number of medical entomology experts only. Authors may consider transferring the manuscript to more specialized entomological journal (e.g. Med. Vet Entomol. or J. Med. Entomol)**

**Reviewer #2: I have already done so in writing to editor.**

**Reviewer #3: This manuscript from Martins et al about the effects of different rearing substrates on L. longipalpis life-table parameters is well written. The strengths of this research include the evaluation of different parameters such as oviposition, survival of larval and adult stages, susceptibility to Leishmania infection and composition of midgut microbiome.**

**--------------------**

**PLOS authors have the option to publish the peer review history of their article (https://journals.plos.org/plosntds/s/editorial-and-peer-review-process#loc-peer-review-history). If published, this will include your full peer review and any attached files.**

****

**Do you want your identity to be public for this peer review? For information about this choice, including consent withdrawal, please see our Privacy Policy (https://www.plos.org/privacy-policy).**

**Reviewer #1: No**

**Reviewer #2: No**

**Reviewer #3: No**

**Figure Files:**

**While revising your submission, please upload your figure files to the Preflight Analysis and Conversion Engine (PACE) digital diagnostic tool, https://pacev2.apexcovantage.com. PACE helps ensure that figures meet PLOS requirements. To use PACE, you must first register as a user. Then, login and navigate to the UPLOAD tab, where you will find detailed instructions on how to use the tool. If you encounter any issues or have any questions when using PACE, please email us at figures@plos.org.**

** **

**Data Requirements:**

**Please note that, as a condition of publication, PLOS' data policy requires that you make available all data used to draw the conclusions outlined in your manuscript. Data must be deposited in an appropriate repository, included within the body of the manuscript, or uploaded as supporting information. This includes all numerical values that were used to generate graphs, histograms etc.. For an example see here: http://www.plosbiology.org/article/info%3Adoi%2F10.1371%2Fjournal.pbio.1001908#s5.**

** **

**Reproducibility:**

**To enhance the reproducibility of your results, PLOS recommends that you deposit laboratory protocols in protocols.io, where a protocol can be assigned its own identifier (DOI) such that it can be cited independently in the future. For instructions see https://journals.plos.org/plosntds/s/submission-guidelines#loc-methods**

---

## [Editor Report · Decision Letter 1]

3 Dec 2020

Dear Dr Aguiar Martins,

We are pleased to inform you that your manuscript 'Effects of larval rearing substrates on some life-table parameters of Lutzomyia longipalpis sand flies' has been provisionally accepted for publication in PLOS Neglected Tropical Diseases.

Best regards,

Guy Caljon

Associate Editor

Abhay Satoskar

Deputy Editor

---

## [Editor Report · Acceptance letter]

15 Jan 2021

Dear Dr Aguiar Martins,

We are delighted to inform you that your manuscript, "Effects of larval rearing substrates on some life-table parameters of Lutzomyia longipalpis sand flies," has been formally accepted for publication in PLOS Neglected Tropical Diseases.

Best regards,

Shaden Kamhawi

co-Editor-in-Chief

Paul Brindley

co-Editor-in-Chief
